# Mood Disorder Severity and Subtype Classification Using Multimodal Deep Neural Network Models

**DOI:** 10.3390/s24020715

**Published:** 2024-01-22

**Authors:** Joo Hun Yoo, Harim Jeong, Ji Hyun An, Tai-Myoung Chung

**Affiliations:** 1Department of Artificial Intelligence, Sungkyunkwan University, Suwon 16419, Republic of Korea; andrewyoo@g.skku.edu; 2Hippo T&C Inc., Suwon 16419, Republic of Korea; jeonh120@skku.edu; 3Department of Interaction Science, Sungkyunkwan University, Seoul 03063, Republic of Korea; 4Department of Psychiatry, Sungkyunkwan University School of Medicine, Seoul 06351, Republic of Korea; 5Department of Computer Science and Engineering, Sungkyunkwan University, Suwon 16419, Republic of Korea

**Keywords:** multimodal analysis, anxiety disorder, biomarker, bipolar disorder, heart rate variability, major depressive disorder, mood disorder

## Abstract

The subtype diagnosis and severity classification of mood disorder have been made through the judgment of verified assistance tools and psychiatrists. Recently, however, many studies have been conducted using biomarker data collected from subjects to assist in diagnosis, and most studies use heart rate variability (HRV) data collected to understand the balance of the autonomic nervous system on statistical analysis methods to perform classification through statistical analysis. In this research, three mood disorder severity or subtype classification algorithms are presented through multimodal analysis of data on the collected heart-related data variables and hidden features from the variables of time and frequency domain of HRV. Comparing the classification performance of the statistical analysis widely used in existing major depressive disorder (MDD), anxiety disorder (AD), and bipolar disorder (BD) classification studies and the multimodality deep neural network analysis newly proposed in this study, it was confirmed that the severity or subtype classification accuracy performance of each disease improved by 0.118, 0.231, and 0.125 on average. Through the study, it was confirmed that deep learning analysis of biomarker data such as HRV can be applied as a primary identification and diagnosis aid for mental diseases, and that it can help to objectively diagnose psychiatrists in that it can confirm not only the diagnosed disease but also the current mood status.

## 1. Introduction

The incidence of mood disorder is continuously increasing at a faster rate than in the past and is a major cause of lowering the quality of life. According to the diagnostic and statistical manual of mental disorder fifth edition (DSM-5), a state in which there is no control over mood is defined as a mood disorder, including major depressive disorder, bipolar disorder, dysthymia, and cyclothymia. According to statistics released by the WHO, about 3.8% of the world’s population suffers from depression, and about 87% people die each year from suicide accidents accompanied by depression [1]. In addition, more than 40 million people suffer from bipolar disorder, and the anxiety disorder observed along with depressive disorder is experienced by over 300 million people.

For an accurate diagnosis of such a mood disorder, it is necessary to visit a psychiatrist to measure and observe the subject’s medical history, psychiatric evaluation, assessment scales, and biomarkers [2,3]. In diagnostic criteria, biomarker features have held a relatively small portion in decision making for mental disorders. However, as the objective aspects of these biomarkers have garnered attention recently, there is a surge in active research focusing on employing biomarker-based classification models as diagnostic aids for mood disorders [4]. The methods for data analysis predominantly fall into two categories: studies employing statistical analysis and those utilizing deep learning or machine learning techniques.

In the medical field, statistical analysis techniques [5,6] are mainly used to collect decision-making information for diagnosis or to interpret the results of clinical trials. Statistical analysis techniques mainly determine whether the difference between the treatment groups is significant or not based on the distributional differences. It has an advantage in that it is possible to grasp the correlation between data variables and the effectiveness of individual features in the analysis process and to determine which variables are most helpful in determining disease through repeated training based on the population distribution. However, the rise of big data collection has led deep learning and machine learning algorithms to rapidly develop as an effective solution, and research to find the importance of hidden features by training the deep neural network structure is attracting attention even for variables that do not have statistical significance. Various medical studies have proven the possibility of development into clinical decision support systems (CDSS) through the introduction of new algorithms.

Studies using deep learning technology, including CDSS, utilize various data collected from medical institutions for analysis. Data and key features that can most efficiently learn the characteristics of diseases to be analyzed from studies using existing statistical techniques are used as input data, and diagnosis has been classified and predicted with an algorithm suitable for the data structure. Basically, medical history and demographic information that can identify the subject’s external factors are collected, medical imaging data for lesion areas are used as a representative input dataset, and for mental illness, research was conducted to help decision making through various physiological data.

In this research, HRV and heart-related biomarker data are applied to the deep neural network structure to perform severity diagnosis and subtype classification of mood disorder. Among the diseases covered in the study, major depressive disorder and anxiety disorder are classified into normal, mild, modulate, and severe groups based on the biomarker of the subject. In addition, bipolar disorder classifies whether the subject is bipolar type 1 or type 2. The content of this study is as follows:Improve mood disorder diagnostic performance with a multimodal analysis algorithm.Classify the difference of severity and subtype within the same mood disorder by deep learning technique.Reduce healthcare costs and improve efficiency by proving the possibility of primary screening tools even within limited biomarker data.

## 2. Related Works

### 2.1. Mood Disorder Analysis

The mood disorder diagnosis has been conducted mainly through clinical examinations, medical history of patients, and consultation with medical staff. However, it has become possible to collect various biological data through non-invasive medical devices, and various learning algorithms through machine learning techniques have led to the design and development of efficient analysis systems.

The data used in the studies of the diagnosis of major depressive disorder, one of the representative diseases of mood disorder, include magnetic resistance imaging (MRI) [7,8], functional near-infrared spectroscopy (fNIRS), which is brain image data, and additional data such as electrocardiogram (ECG) and heart rate variability (HRV) that can identify the imbalance of the autonomic nervous system are used a lot. The overall biological information collected for the body’s disease and pathological judgment is called biomarker data [9], and it is also used as a crucial indicator to determine the existence, severity, or treatment response of physical or mental diseases according to their representative characteristics.

Heart rate variability (HRV) is an index that measures the change in minute time intervals between cycles of heartbeat and has been used to identify variability and check trends in physical diseases [10]. Based on these characteristics, many studies have been conducted to identify and predict changes in myocardial infarction [11], coronary artery disease [12], and cardiovascular diseases [13]. However, in the field of psychophysiology, HRV measurement is used as a non-invasive tool to study autonomic nerve function, and autonomic nervous system dysfunction is used in screening tools that can discriminate anxiety disorders and major depressive disorders, especially in that it adversely affects many psychiatric disorders [14,15].

In studies of depressive disorders, MRI data were mainly used to confirm the etiology and pathogenesis of depression [16]. In Marzieh Mousavian’s research [17], these structural and functional MRI data were used to extract major features and prove the best performance in the Bernoulli Naive Bayes classifier. In a study using fNIRS data [18], which measures the degree of oxyhemoglobin and deoxyhemoglobin in the brain bloodstream, the author showed that major depressive disorders and normal controls were classified with about 99.94% accuracy using vector-based features. In addition, the difference in HRV parameters between the health control group and the depression group was significant when antidepressants were prescribed for 2 weeks, proving the correlation with depression [19]. Finally, in the depressive disorder severity classification study conducted by our team [20], a federated learning-enhanced model with an accuracy of about 90.7% was presented using HRV variables.

Another disease associated with major depressive disorders is anxiety disorder, and biometric data similar to the aforementioned studies are used in the research area. In a study classifying generalized anxiety disorder (GAD) from major depressive disorder [21], the multimodal classification accuracy of 90.10% was confirmed through the binary support vector machine using clinical questionnaires, cortisol release, gray matter, and white matter volume data variables as input values. In addition, in a study based on the relationship between mental illness and abnormal communication in the brain areas [22], a machine learning model was proposed to classify generalized anxiety disorder using electroencephalogram (EEG) data. It showed a classification performance of 97.83% through the extraction of multidimensional features of EEG and the neural network-based bagging strategy. However, the above studies have a limitation in that they have not been able to analyze the severity of subjects’ anxiety disorder, since they are mainly used for binary classification of whether the subject has an anxiety disorder or not.

Finally, bipolar disorder (BD) is a disease with a wide range of mood changes with hypomania, mania, and remission conditions. There are also studies that applied biomarker datasets for diagnosing bipolar disorder [23,24], but most of the existing studies have used an audio/visual emotion challenge and workshop dataset (AVEC) for the diagnosis and classification of bipolar disorder. In multimodal ensemble research [25], the speech signal was mainly used among the data variables collected from the AVEC data. The clinical state of bipolar disorder patients is classified by analyzing the treatment of mel-frequency cepstral coefficients (MFCC) and geneva minimalistic acoustic parameter set (GeMAPS) with a multimodal classifier consisting of convolutional neural networks and multi-layer perceptron layers. The hybrid model classification research [26], another study for the same dataset, extracted the facial features with the CNN model, and the long short-term memory (LSTM) model was added to classify the three different states of BD. Existing studies have generally used visual/audio datasets for the classification of bipolar disorder patients’ states, but it is challenging to apply to the medical environment due to data privacy issues. Additionally, they have limitations in that it is classified into three states rather than the subject’s subtype of bipolar disorder.

### 2.2. Multimodal Analysis on Medical Dataset

Multimodal analysis is proposed as a method that can more efficiently analyze datasets with multiple characteristics at once [27,28]. In particular, in the field of medical artificial intelligence, various data that can be collected from one patient are utilized to make decisions on improved performance through the main characteristics of each data. Data with different structures and characteristics are largely divided into feature concatenation and ensemble classifiers for multimodality analysis. Feature concatenation is used to extract the main characteristic vectors from each data and analyze them in one integrated vector form, while the ensemble classifier is a technique that performs majority voting based on the results analyzed by individual models.

In the field of medical AI research, multimodal analysis is particularly used to improve the performance of diagnosis based on medical imaging data [29,30]. Various medical images are taken from one subject to diagnose the disease, and the characteristics of the data shown by each medical device are different. In other words, even if photographing in the same area is performed, oxygen activation can be shown depending on the image or the lesion area that can be found, so various data from the same area can be trained together to increase the accuracy of decision making. In the deep learning multimodal medical imaging research [31], Zhe Guo proved that the multimodality analysis of PET (Positron Emission Tomography), CT (Computed Tomography), and MRI (Magnetic Resonance Imaging) images taken in the same environment was performed to improve the accuracy of segmentation techniques to find disease areas, and improved performance compared to single-modality was demonstrated.

## 3. Mood Disorder Classification

### 3.1. Dataset

In this study, a physiological dataset of 3687 patients collected from the Samsung Medical Center (SMC) was used to perform mood disorder diagnosis and subtype classification. These are heart rate variability test data from patients aged 20 to 60 years who were conducted from 1 January 2016 to 15 June 2022 at the depression center in the department of psychiatry. The data of each subject, including the physiological dataset, includes demographic information, HRV indices, medical history, and assessment scale results including the Hamilton Depression Rating Scale (HAMD), Hamilton Anxiety Rating Scale (HAMA), Beck Depression Inventory (BDI), Beck Anxiety Inventory (BAI), Mood Disorder Questionnaire (MDQ), and Hypomania Symptom Checklist-32 (HCL-32). When selecting the data, heart diseases such as arrhythmia and heart failure, and heart disease-related procedures such as artificial heartbeat insertion high-frequency electrode catheter resection, or thoracic surgery were excluded. The data used in this study were selected and used for analysis using previously recorded data not newly collected data for research. All processes were carried out under Institutional Review Board (IRB) approval.

Since our team selected major depressive disorder and anxiety disorder for severity diagnosis, and the mood disorder subtype diagnosis is about bipolar disorder, a separate preprocessing process for each task was performed first. In addition, among the collected assessment scales, HAMD, HAMA, BDI, and BAI scores were used for data preprocessing and severity labeling.

### 3.2. Data Preprocessing

First of all, data preprocessing for mood disorder severity diagnosis was carried out based on assessment scales for each disorder as shown in Figure 1. Most of the existing studies use the diagnosed disorder class from the medical center as a classification label, but this study attempted to create a more objective indicator of severity classification model by using the collected assistance scale results, which are tools that measure the subject’s physiological state at the time as a training dataset. Diagnostic results made by psychiatrists based on various indicators such as the subject’s medical history, biomarker, stress level, and demographic information exist in the data, but they only have the type of disease without information on severity. Therefore, severity labeling was performed using the results of the verified assessment scale conducted under the structured interview with psychiatrists not the self-assessment tools.

The first step commonly processed in data preprocessing for mood disorder severity and subtype classification is to filter the noise data due to errors in HRV measurement. In order to exclude this from the training and validation set, two outlier criteria were set to process the entire dataset. All data with a successive RRI difference (SRD) of less than 0.8 or more than 1.0 among HRV indicators, or data with an abnormal heart rate of more than 5 for identifying abnormal conditions during heart rate measurement, were determined to have had a measurement error and removed from the entire data.

The Hamilton Depression Rating Scale (HAMD) and Beck Depression Inventory (BDI) are used simultaneously among the assistance scale results to classify the severity group of mood disorder. HAMD is classified by the severity of normal, mild, moderate, or severe based on 7, 18, and 25 points, and BDI is also classified into four severity categories based on 14, 19, and 28 points. As a classification target label for this study, a new column was generated and used for the case where the results of the above two assistance scales used for depression measurement matched. For example, if the subject was classified as a severe depressive disorder through the HAMD scale, and the result of BDI was also classified as a severe depressive disorder, it was newly designated as a “severe” group.

The severity group classification process of the anxiety disorder was also similar to that of the major depressive disorder. Here, the results of the Hamilton Anxiety Rating Scale (HAMA) and Beck Anxiety Inventory (BAI), which can check the subject’s degree of anxiety status, were used as an assessment scale for classification. HAMA proceeds with four classifications based on 18, 25, and 30 points, and BAI classifies the severity based on 8, 16, and 26 points. Similarly, for secondary classification through two assistance scale results, a new column called “anxiety” was created for users who output the same results with each scale and designated as an anxiety disorder severity classification target.

Finally, bipolar disorder attempted to create a classification model that distinguishes the subtype of the disease rather than determining the severity of the disease. Therefore, the group was not newly classified through a separate assessment scale but was divided into bipolar disorder type 1 and bipolar disorder type 2 through main dx (main disorder) diagnosed by SMC psychiatrists.

Table 1 shows the data information of each classification task after the preprocessing process. The average age of the subjects with major depressive disorder and anxiety disorder is 40 to 41 years old, and the average age of the subjects used in the subtype classification of bipolar disorder is 30 to 31 years old, which is 10 years lower than this. In addition, it can be seen that the proportion of women is generally high, with 61.4 to 38.6 in major depressive disorders, 57.1 to 42.9 in anxiety disorders, and 67.5 to 32.5 in bipolar disorders. In addition, statistics on the Hamilton Depression Rating Scale and Beck Depression Inventory for depression measurements and the Hamilton Anxiety Rating Scale and Beck Anxiety Inventory for anxiousness measurements are also available in the table.

Checking the preprocessing process, it can be seen that the preprocessing of the bipolar disorder, MDD, and AD groups is different. In the collected data, the main, second, and third diagnosis results of each subject evaluated by SMC psychiatrists are included in the medical history, so disease groups can be easily classified. However, the data mainly used in this study is heart rate variability (HRV), which is the biomarker data measured when visiting to collect the data. In other words, it was determined that the subject’s previously diagnosed dx result was not the result of the clear state at the time of HRV measurement. Therefore, for the MDD and AD groups classification, the subjects’ target group is required to be newly defined to represent the state at the time of measurement using the results of the assessment tool.

### 3.3. Severity and Subtype Diagnosis Model

In this study, a deep neural network-based system was designed and proposed for mood disorder severity diagnosis and subtype classification. In existing medical diagnostic studies, a method of analyzing statistical significance with the comparison group was generally selected, or decision making was made through algorithms such as support vector machines. However, when statistical analysis was attempted prior to the application of the proposed deep neural network structure, it was found that there was a limit to performing analysis of mental illness due to the statistical significance between specific variables. In other words, for a complex classification such as disease severity and subtype, it is necessary to grasp the hidden feature of the data, and for this, the multimodality deep neural network algorithm was judged to be appropriate.

Existing studies have confirmed the relationship between mental health and HRV, and the possibility as a diagnostic tool has been confirmed because the features extracted from the time and frequency domain reflect the state of the autonomic nerve system. However, it showed limitations as an objective indicator due to the cause of disorder, duration, lifestyle, or even individual’s stress level were reflected in HRV data. Prior to the reflection of the proposed algorithm, the results shown in Table 2 below were obtained when checking which variables actually had statistically significant differences between the severity or subtype of mental illness.

The collected HRV dataset can be classified into time domain method or frequency domain method. Among HRV variables, the standard deviation of NN interval (SDNN), square root of the mean of the sum of the square of differences between adjacent NN interval (RMSSD), approximate entropy (ApEn) are classified into time-domain method, and low frequency (LF), high frequency (HF), very low frequency (VLF), very high frequency (VHF), high and low-frequency ratio, and total power (TP) are classified into frequency domain methods. Frequency domain variables mainly decompose signals from time-series data and are indicators of the balance between sympathetic and parasympathetic nerves. On the other hand, time-domain variables are used to check signal complexity, activity of sympathetic and parasympathetic nerves, and physical stress index.

According to the statistical significance test under *p*-value of 0.05, there are only two significant variables in the severity group comparison of MDD, zero significant variables in the comparison of AD, and five significant variables in the subtype comparison of BP. In other words, it can be seen that the limitations of the statistical approach are clear for severity or disease subtype recognition beyond simple disease classification. This is because although it is possible to accurately classify the group only when there is a clear difference in the core data variables of HRV representing the autonomic nerve system, there is an insignificant difference in the variables when looking at the results of statistical analysis. Therefore, we adopted the deep neural network algorithm under the assumption that data variables without statistical significance would have hidden features to help classify severity or subtype. The overall system of the proposed architecture is shown in Figure 2.

As described in the previous section, the data used in the study are medical history and biomarker data of the group diagnosed with mood disorder, and they were applied to the classification model after passing through the individual preprocessing step designed for each data characteristic. Specifically, out of all data collected from individual subjects, a total of three variable groups were used as input data: HRV in time-domain, HRV in frequency-domain, and Autonomic Nerve System-related data. There is also a method of analyzing data with three different characteristics through a separate model and making a decision through major voting, but multimodal analysis was performed by referring to the use of one concatenated biomarker feature vector from the results of previous studies. For integration into concatenated single vector dimensions, training data were configured only with core variables that removed duplicate data such as natural logarithm value from each data.

Before applying the data to a classification model composed of fully connected layers, we tried to solve the insufficient medical data issue by adding a 5-fold cross-validation step. As an example, the four severity groups of the major depressive disorder have data of 122, 189, 145, and 143 people, respectively, which is actually insufficient for training using deep learning. Therefore, k-fold cross-validation was used to perform repetitive learning by dividing the entire data into k sets and k-1 parts into training data and the remaining one part into test data without dividing the entire dataset into train, test, or validation at once. During the experiment, the optimal k value was determined as 5, so the 5-fold cross-validation was applied to solve the lack of data issue.

A deep neural network structure was created by placing the 5-fold cross-validation as the base architecture and attaching a fully connected layer. The concatenated vector input has 17 variables, and it is configured to learn through a total of six dense layers. With a preprocessed dataset, each concatenated feature vector goes through a multimodal deep-learning structure as shown in the Table 3. Major depressive disorders and anxiety disorders have four target severity labels, and bipolar disorders have two target subtype labels, so the structure of the last dense layer changes accordingly and the rest of the structures are identical. In addition, batch normalization layers were added between the dense layers to prevent being stuck at the local optimum value. The results of mood disorder severity and subtype classification applying a test dataset to the trained model are as follows.

In this study, four indicators were used to confirm the performance of the experimental results as shown in Figure 3: precision, recall, accuracy, and F1 score. These are indicators that are widely used in diagnostic research, and it is possible to compare the predicted results with the actual results and also evaluate the validity of the model. For the three different classification tasks covered in the study, a classification model was generated as described above, and the result of analyzing the test dataset using this is shown as a confusion matrix. Using the corresponding results, it is possible to calculate the precision, recall, accuracy, and F1 score indicators for performance confirmation.

First, the results for MDD among the mood disorder severity classification tasks are shown in Figure 4. The labels 0, 1, 2, and 3 of the confusion matrix represent normal, mild, moderate, and severe depressive disorder groups. The analysis results for the four severity categories of major depressive disorder are 0.8833 of the average classification accuracy, 0.8862 precision score, 0.8833 recall score, and 0.8840 F1-score.

In addition, the results of the severity analysis for subjects with anxiety disorder are shown in Figure 5. Similar to the classification of major depressive disorder, 0, 1, 2, 3 marks of the confusion matrix in the figure represent normal, mild, moderate, and severe anxiety disorder groups, respectively. The analysis results of the four severity categories of anxiety disorder are 0.8725 of the average classification result, 0.8462 of the precision score, 0.8724 of the recall score, and 0.8713 of the F1 score, respectively.

Finally, Figure 6 is the classification result of what subtype is in the group diagnosed as bipolar disorder. On the confusion matrix, 0 is type 1 bipolar disorder and 1 is type 2 bipolar disorder. The binary classification for the bipolar disorder subtype classification results are 0.8333 of the average classification accuracy, 0.7666 of the precision score, 0.8333 of the recall score, and 0.7976 of the F1 score, respectively. The bipolar disorder subtype classification shows relatively lower performance than the severity classification for the other two mood disorders, which is confirmed as a result of less data and severe imbalance between subtypes compared to other tasks.

As described above, existing medical data studies have mainly applied statistical analysis techniques, and they used simple machine learning algorithms such as support vector machine (SVM) for classification. Therefore, to check the performance of the classification algorithm that our team proposed, the performance was compared by applying the SVM and support vector machine recursive feature elimination (SVM-RFE) methods to the same medical data. The reason for the additional application of SVM-RFE is that it was confirmed through an ANOVA experiment that the statistically significant difference between the data variables used in this study was not large, and it was thought that performance improvement could be achieved if variables with little effectiveness were removed when applying SVM. The performance comparison of the three mood disorder classification tasks conducted in the experiment is shown in Table 4.

The average classification accuracy using the statistical approach was 0.752 in SVM and 0.765 in SVM-RFE. The accuracy was 0.883 when the proposed multimodal DNN was applied for the four severity classifications of major depressive disorder, which is a 0.118 accuracy increase compared to the SVM-RFE result. For the severity classification of anxiety disorder, the classification accuracy of 0.625, 0.642, and 0.873 were shown, respectively, and the performance improved by about 0.231 using the proposed algorithm. Finally, for the subtype classification of bipolar disorder, the classification accuracy of 0.708, 0.708, and 0.833 are shown, and the classification accuracy performance improvement can be confirmed by about 0.125.

## 4. Discussion

Deep learning and machine learning algorithms have introduced medical systems that help physicians make decisions. Most of the medical AI research plays an auxiliary role in diagnosing physical diseases from medical imaging data such as MRI and CT. Some studies collect biomarker data of patients and adopt it to determine the presence of mental illness, but the objective criteria for accurate diagnosis have not yet been established and have not shown sufficient performance for diagnosis. In addition, there are issues that make it more challenging to judge the severity or subtype of the disease classification within the same disease.

This study performs the classification of the severity and subtype of mood disorder by applying a multimodal deep learning algorithm to heart rate variability data and autonomic nerve system-related data variables. The analysis result was compared with representative statistical analysis techniques used in the existing medical AI research to evaluate the classification performance. The statistical approach is based on the method of determining whether there is a significant difference through the distribution of data variables between each class to be classified. However, we determined that there would be a difference between the data variables of the biomarker that could be trained from the hidden feature in addition to the distributional comparison. Therefore, a deep neural network was adopted for its ability to train complex hidden features through a neural network layer.

In addition, multimodal analysis was used because it showed improved classification accuracy than single modality studies that learned data of one characteristic. In our previous study, functional near infrared spectroscopy (fNIRS) data and verbal fluency test (VFT) data were used to determine major depressive disorders and suicidal tendencies. When comparing the result of single modality analysis using each data variable to multimodal analysis using two data variables as concatenated feature vectors, we confirmed an average improvement in sensitivity of 0.107 and a specificity improvement of 0.157. As a result, the advantages of the multimodal analysis algorithm were confirmed through the study, so it was also introduced in this study.

As a result, the proposed multimodal deep neural network model using both the time-domain and frequency-domain variables of HRV data and ANS variables resulted in a performance improvement of 11.8% in the diagnosis of major depressive disorder severity, 23.1% in the diagnosis of severity of anxiety disorder, and 12.5% in the diagnosis of bipolar disorder subtype. Although it still does not have high accuracy enough to completely replace the existing diagnostic method, it can be confirmed that it shows high performance even though multiclass classification is performed using limited data features.

Our research team thinks the results of this study can be of great help to the mental health field in the Republic of Korea. In 2022, the seriousness of mental health is increasing to the extent that about 440k people in Korea suffer from depression above moderate severity, but the relative lack of infrastructure for efficient diagnosis and treatment is emerging as a continuous social issue. Since it is important to detect mental diseases such as depression and anxiety disorders early on and provide steady management, it is expected that it will become an efficient screening tool for building mental health infrastructure if data can be collected with non-invasive diagnostic tools such as HRV and the severity can be confirmed through deep learning algorithms as suggested in this study. It is also expected that much more efficient and accurate discrimination will be possible if people with high severity or risk are identified after being used as a primary screening tool and clinical diagnosis.

### Limitations

The main limitation of this study is still the lack of medical data. The initial amount of data used in the study was about 3600 subjects’ data, but after preprocessing such as noise filtering and outlier removal, the number of data required for training and verification was significantly reduced as the extraction of disease groups required for analysis tasks was performed. This means that there is not enough training data to apply the deep neural network, and even if k-fold cross-validation was applied, characteristic training such as overfitting for some classes may not be sufficient due to the imbalance data issue by severity groups.

Therefore, future studies will improve the deep neural network to be utilized under high performance even in an insufficient data environment by using data augmentation that can be applied to the medical data environment using the same data used in this study.

## 5. Conclusions

Three types of mood disorder analysis were conducted that are difficult to classify using biometric signal data collected in hospitals. For the major depressive disorder group and anxiety disorder group, classification was performed into four groups: normal, mild, moderate, and severe so that the current severity status could be identified, in addition to discriminating the difference in the presence or absence of simple diseases. For bipolar disorder, a subtype classification was conducted to distinguish between type 1 and type 2 bipolar disorder. In existing medical studies, disease discrimination was performed by comparing the validity and checking the distributional difference between groups of medical data variables, but the deep neural network structure was applied to find the hidden features between variables. Finally, the performance was further improved by performing feature extraction from HRV’s time-domain and frequency-domain and then performing multimodal analysis by extracting additional features from ANS variables.

Through the research results, our research team confirmed that mood disorder severity and subtype classification are possible by utilizing heart-related data without collecting complex data such as audio or visual signals. Through this, we expect that it will be used as an auxiliary tool for the diagnosis and determination of mental diseases in medical institutions in the future.

## Figures and Tables

**Figure 1 sensors-24-00715-f001:**
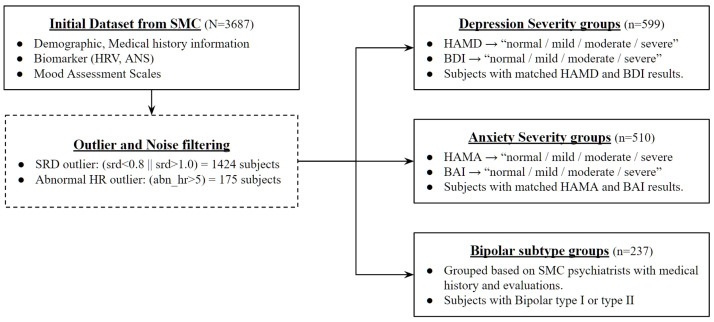
Data preprocessing steps of overall dataset.

**Figure 2 sensors-24-00715-f002:**
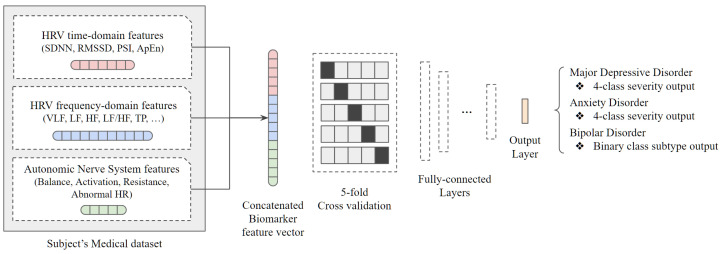
Overall architecture to diagnose severity or subtype of the mood disorders using multimodal deep neural network model.

**Figure 3 sensors-24-00715-f003:**
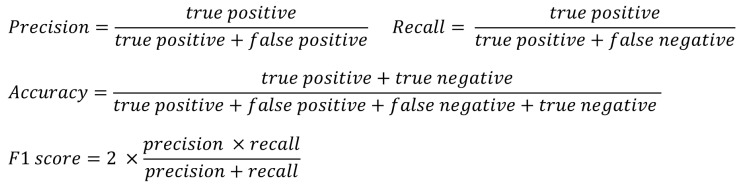
Equations for calculating classification accuracy, precision, recall, and F1 score.

**Figure 4 sensors-24-00715-f004:**
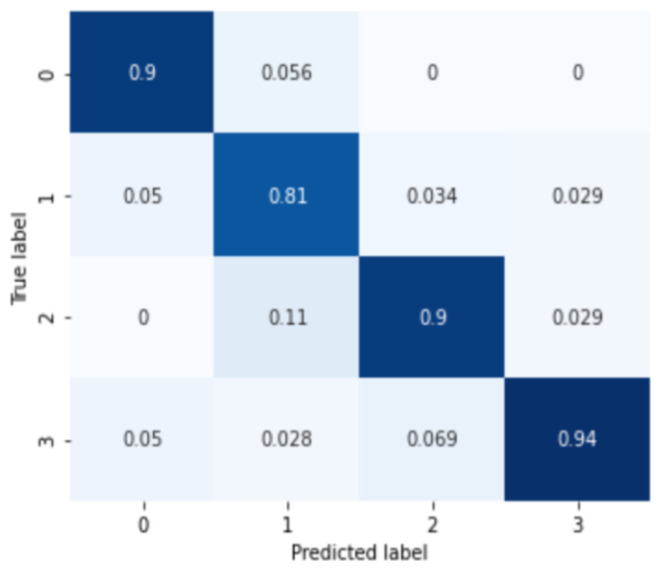
Confusion matrix of major depressive disorder severity classification.

**Figure 5 sensors-24-00715-f005:**
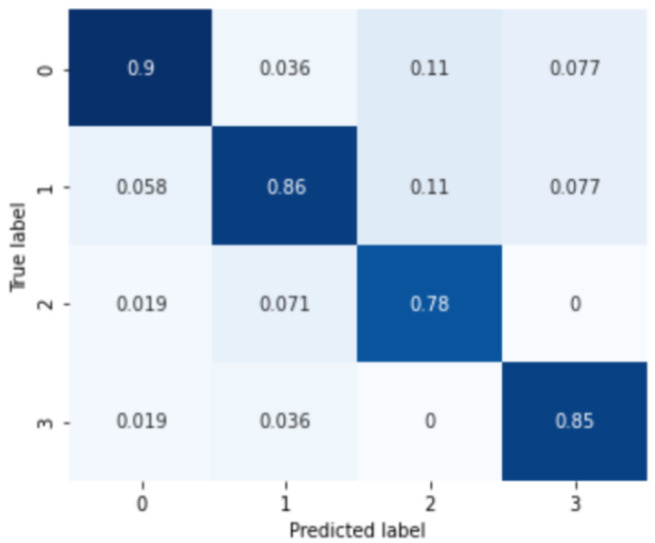
Confusion matrix of anxiety disorder severity classification.

**Figure 6 sensors-24-00715-f006:**
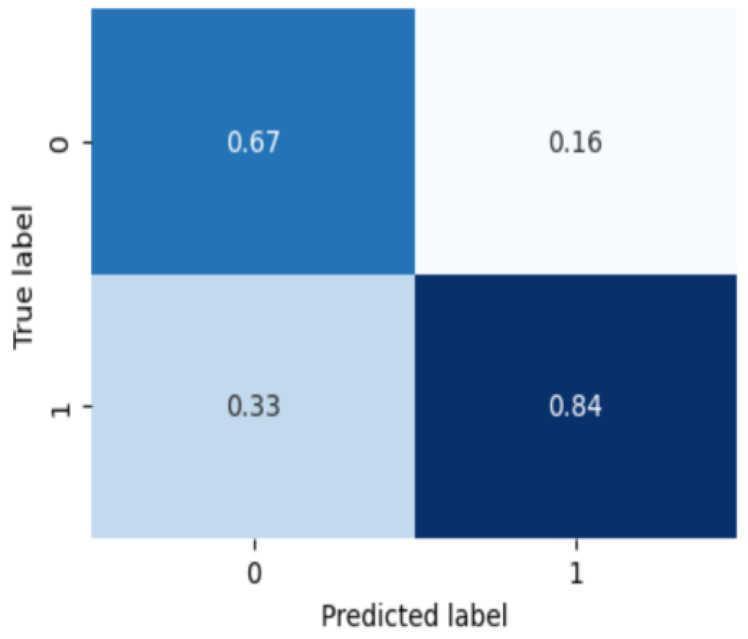
Confusion matrix of bipolar disorder severity classification.

**Table 1 sensors-24-00715-t001:** Summary of the each classification dataset.

	MDD Subjects (*n* = 599)	AD Subjects (*n* = 510)	BD Subjects (*n* = 237)
Age	41.81 ± 13.95	40.99 ± 14.11	30.80 ± 10.55
Gender	F (368), M (231)	F (291), M (219)	F (160), M (77)
HAMD	16.13 ± 8.57	14.71 ± 7.44	16.79 ± 6.96
HAMA	17.51 ± 8.29	17.24 ± 12.67	17.22 ± 7.76
BDI	22.29 ± 13.80	20.37 ± 14.24	32.91 ± 13.65
BAI	19.65 ± 14.30	12.44 ± 13.43	23.94 ± 14.13

**Table 2 sensors-24-00715-t002:** ANOVA statistical analysis with HRV data variables on three different classification tasks.

	Major Depressive Disorder (MDD)		Anxiety Disorder (AD)		Bipolar Disorder (BD)	
**HRV Features**	* **F** * **-Value**	* **p** * **-Value**	* **F** * **-Value**	* **p** * **-Value**	* **F** * **-Value**	* **p** * **-Value**
SDNN	1.83785	0.13907	0.26939	0.84747	7.35061	**0.00719**
RMSSD	0.69511	0.55526	0.36338	0.77947	2.79095	0.09612
ApEN	2.83537	**0.03661**	0.96904	0.40703	0.34025	0.56024
TP	1.91545	0.12586	0.81340	0.48683	3.29967	0.07056
VLF	1.61488	0.18474	0.12525	0.94515	4.77567	**0.02985**
LF	2.12205	0.09629	1.00413	0.39063	0.71508	0.39862
HF	0.41216	0.74432	1.19709	0.31026	1.26335	0.26216
LF/HF	0.79464	0.49717	1.60874	0.18640	2.40641	0.12218
LF norm	1.66954	0.17239	1.53822	0.20368	0.03966	0.84231
HF norm	1.66955	0.17239	1.53822	0.20368	0.03966	0.84231
SRD	0.70152	0.55138	0.73285	0.53274	0.02533	0.87366
TSRD	0.58352	0.62599	0.37995	0.76749	3.09363	0.07990
ln(TP)	2.13602	0.09455	0.31154	0.81704	11.09756	**0.00101**
ln(VLF)	2.01519	0.11063	0.17302	0.91462	16.83902	**0.00005**
ln(LF)	2.86199	**0.03621**	0.47418	0.70039	5.30771	**0.02211**
ln(HF)	0.45809	0.71167	0.56139	0.64071	5.90915	**0.01581**

**Table 3 sensors-24-00715-t003:** Deep neural network structure for mood disorder severity or subtype classification.

Layer (Type)	Output Shape	Param Count
dense_1 (Dense)	(None, 64)	1152
dense_2 (Dense)	(None, 256)	16,640
batch_normalization_1 (BatchNormalization)	(None, 256)	1024
activation_1 (Activation)	(None, 256)	0
dense_3 (Dense)	(None, 256)	65,792
batch_normalization_2 (BatchNormalization)	(None, 256)	1024
activation_2 (Activation)	(None, 256)	0
dense_4 (Dense)	(None, 256)	65,792
dense_5 (Dense)	(None, 64)	16,448
dense_6 (Dense)	(None, class number)	260

**Table 4 sensors-24-00715-t004:** Average classification accuracy comparison between SVM, SVM-RFE, and proposed DNN method.

	SVM	SVM-RFE	DNN	Diff
Major Depressive Disorder	0.752	0.765	0.883	+0.118
Anxiety Disorder	0.625	0.642	0.873	+0.231
Bipolar Disorder	0.708	0.708	0.833	+0.125

## Data Availability

Data are unavailable due to privacy or ethical restrictions.

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
