# Peer review of "Mood Disorder Severity and Subtype Classification Using Multimodal Deep Neural Network Models"

_sensors, 2024, doi:10.3390/s24020715_

Round 1

Reviewer 1 Report

Comments and Suggestions for Authors

Hello Dears;

Thanks for the research.

Comments:

-Machin learning and deep learning are novel fields of research, that are one of the strengths of this article.

-Psychiatric disorders are multi factorial and it is not possible to diagnose or determine the severity of the disorders only with HRV or imbalance of the autonomic system .

-The neurotransmitters associated with BD, AD and MDD are diverse and not only limited to the imbalance of the autonomic system, and serotonin, dopamine, glutamate etc. are also involved, and they are all bio-psycho-social disorders.

- In scientific research ,the diagnosis of psychiatric disorders is not enough with the diagnosis of one psychiatrist, and it should be confirmed with standard questionnaires or semi-structured interviews.

Author Response

Dear, Sensors reviewers.

First of all, thank you for suggesting us to submit revised draft of our manuscript entitled, "Mood disorder severity and subtype classification using Multimodal Deep Neural Network models" to Sensors. We also appreciate the time and effort you have dedicated to providing insightful feedback on ways to strengthen our paper. We thought it was a paper that lacked a lot, so our research team revised the entire manuscript to make a better paper. We have incorporated changes that reflect the detailed suggestions you have graciously provided. We also hope that our edits and the responses below satisfactorily address all the issues and concerns you have noted.

Response to Reviewer #1

Point 1. Classification of severity with HRV or imbalance of the autonomic system.

Response: Our research team agrees with the opinion that there must be better explanations on how it is possible to determine the severity of mood disorders using HRV or imbalance of the autonomic systems. Therefore, we added detailed explanations of the heart rate variability and other related datasets to the manuscript. We also add the significance of the dataset for the mental disorder binary classification or severity classification tasks. You can find the added contents on the pages 2 to 3 with the red text color.

Point 2. Explanations on the rating scales and dataset.

Response: Our research team agrees with the opinion that there must be better explanations on the adopted rating scales for the data labeling. Basically, the dataset used for the study contains assessment scales including HAMD, HAMA, BDI, BAI, MDQ, and HCL-32.  However, the measures for depression and anxiety disorders focused on this study were HAMD, HAMA, BDI, and BAI, so they were used. Especially, HAMD and HAMA assessment tools are processed under the structured interview with psychiatrists. In addition, the reason why the labeling was renewed using the scale rather than the result diagnosed by the medical staff is that the HRV data could be a clear basis for determining the subject's condition at the time of measurement. You can find the added contents on the pages 4 with the red text color.

Reviewer 2 Report

Comments and Suggestions for Authors

This is a review of a manuscript titled: “Mood Disorder severity and subtype classification using Multimodal Deep Neural Network models”. This is an important work considering the usual techniques making diagnoses about mood disorders, there hasn't been enough research that considers HRV as a biomarker which provides interesting insight about the topic. 

Line 11. Please give more context for a better general understanding of the results of accuracy performance. 

Line 27. Expand on the characteristics of the population of South Korea where you obtained your data. 

Line 31. Could you include what is a biomarker and why it is important?

Line 39. Please include a couple of examples of the analysis techniques and mood disorders biomarkers. 

About the sample, it is important that you include how the data were collected and if they signed an ethical concern form. 

Line 147. Please describe the characteristics of the medical center. Include demographics about the group (although some of them are presented in Table 1). 

Line 168. In introduction, you need to expand about previous related work using those scales. 

Table 2. For a better understanding of the HRV features, include a descriptive paragraph about the relevance of the variables and what do they stand for. 

 Authors should deploy how the deep neural network was constructed. It can be published as an open file on an outside storage site. 

Discussion. It is important to discuss the results you have and what the results tell about the mood disorders in your population. It seems limited about mentioning only the importance of deep learning, but the general results are not described. 

Best regards, 

Author Response

Dear, Sensors reviewers.

First of all, thank you for suggesting us to submit revised draft of our manuscript entitled, "Mood disorder severity and subtype classification using Multimodal Deep Neural Network models" to Sensors. We also appreciate the time and effort you have dedicated to providing insightful feedback on ways to strengthen our paper. We thought it was a paper that lacked a lot, so our research team revised the entire manuscript to make a better paper. We have incorporated changes that reflect the detailed suggestions you have graciously provided. We also hope that our edits and the responses below satisfactorily address all the issues and concerns you have noted.

Response to Reviewer #2

Point 1. Explanations on accuracy performance.

Response: Our research team agrees with the opinion that there must be better explanations on experiment performance calculation. In this study, a classification study was conducted on mood disorders, and a total of four indicators were used to confirm the research results: precision, recall, accountability, and f1 score. After classification is performed using the test dataset, the confusion matrix is generated as a result, which are the four indicators that can be calculated based on this. This is widely used in medical research and machine learning research, and a method for calculation was added to the manuscript. You can find the added contents on the pages 8 with the red text color.

Point 2. Explanations on dataset characteristics.

Response: Our research team agrees with the opinion that there must be better explanations on the adopted dataset. The data used in this study was previously collected for mental health research by SMC, and our existing manuscript was missing the detailed data description. It is data collected for subjects between the ages of 20 and 60 and includes biomarker data sets including HRV, results from various rating scales, and results for diagnosed mental illness. This part was newly added to the text. You can find the added contents on the pages 4 with the red text color.

Point 3. Explanations on biomarker dataset.

Response: Our research team agrees that there must be better explanations of the biomarker dataset and its significance. Biomarker data is information that helps physicians to determine about body’s disease or pathological judgments. In addition to obtaining additional information about the disease, the biomarker data is incorporated into machine learning algorithms and has developed to the level of being able to diagnose various diseases. Against this background, this research team conducted a mood disorder study using a biomarker called HRV as the main. This part was newly added to the text. You can find the added contents on the pages 2 to 3 with the red text color.

Point 4. Additional reference search for biomarker research.

Response: Our research team agrees that there must be better evidence about the biomarker research and its significance. Therefore, along with a detailed explanation of HRV, the main used biomarker dataset, an additional investigation was conducted on how the data characteristics of HRV could be used in the mood disorder classification study. You can find the added contents on the page 3 with the red text color.

Point 5. Ethical concern on the dataset.

Response: Our research team agrees that there could be concern on our dataset with the ethical issue. All the data used in this study were collected at the depression center of the Samsung Medical Center, and both research design and IRB application and approval processes were carried out prior to the study. Data disclosure is impossible because it can contain sensitive information about individual patients, but information on IRB approval will be provided to the journal.

Point 6. Explanations on HRV variables.

Response: Our research team agrees that there must be better explanations on HRV variables used in the study. As mainly used in HRV analysis, this research team used variables by dividing them into the time domain and frequency domain, and each variable has various information such as the balance between sympathetic and parasympathetic nerves, activity level, or even physical stress level. Although the possibility was confirmed through statistical analysis in advance, there were not enough results to be judged by statistical analysis, and the background of using the deep neural network was explained accordingly. You can find the added contents on the page 7 with the red text color.

Point 7. Deep Neural Network model.

Response: Our research team agrees that there must be better explanations of deep neural networks model adopted for the study. Our research team agrees that there must be better explanations of the deep neural networks model adopted for the study. This study used a deep neural network model for final test dataset inference. Although there are various pre-trained models provided by Tensorflow or Pytorch, we made various attempts to create a model that can learn the characteristics of HRV data used by the research team well, and as a result, a model structure such as Table 3 with a total of 6 dense layers and a batch-normalization added between them was selected. Only the output size of the dense layer for the last decision was changed and used according to the classification task. You can find the added contents on the page 8 with the red text color.

Point 8. Discussion section.

Response: Our research team agrees that there must be better explanations of the result of the study, and how this research can be meaningful conclusion. In the discussion section, the current situation in related fields in Korea and how these research results can be helpful are newly described. Currently, the severity of mental illness continues to increase after the pandemic, and the infrastructure to solve it is insufficient. Therefore, it would be helpful if it could be used as a screening tool through simple data as suggested in the results of this study. You can find the added contents on the pages 10 to 11 with the red text color.

Reviewer 3 Report

Comments and Suggestions for Authors

This is a valuable contribution which aspires at delivery of biological marker - based calssification of mood disorders and more specifically, of their severity. The topic is crucial from the viewpoint of medicine, where the grading and staging of illness is fundamental premise for clinical decision-making. In this study there has been employed deep neural network model to classify a retrospective sample by use of the autonomous nervous system measures, such as heart rate variablity. The latter are well known to be associated with mood disorders since the times of classical neuropsychiatry. My recommendation is to better fit the current research into its approipriate context. In the past few years there has been produced a remarkable advance in an attempt to construct neuroscience-informed psychiatric classification, namely the so called nomothetic-networks psychiatry approach. It consolidates various data sources into partial least squares pathways analysis. In turn , the nomothetic networks represent clear associations between molecular, pathophysiologcal, imaging and clinical measures, that may inform and validate diagnostic procedures and the present study is definitely contributing to that approach. 

I would also recommend that the paper is reviewed by an expert in statistical methods. 

Author Response

Dear, Sensors reviewers.

First of all, thank you for suggesting us to submit revised draft of our manuscript entitled, "Mood disorder severity and subtype classification using Multimodal Deep Neural Network models" to Sensors. We also appreciate the time and effort you have dedicated to providing insightful feedback on ways to strengthen our paper. We thought it was a paper that lacked a lot, so our research team revised the entire manuscript to make a better paper. We have incorporated changes that reflect the detailed suggestions you have graciously provided. We also hope that our edits and the responses below satisfactorily address all the issues and concerns you have noted.

Response to Reviewer #3

Response: Our research team revised the overall manuscript to provide more professional information and clarify opinions by reflecting on the various opinions provided by the reviewers as much as possible. Various contents have been added, such as detailed explanations of the dataset, deep learning analysis algorithms, performance evaluation methods, and the meaning of this study in the future. We would really appreciate it if you could review this again.

In addition, statistical analysis was reviewed with the help of Ji Hyun An, a psychiatrist of the Samsung Medical Center, and the results were confirmed and proved based on various research backgrounds conducted in medical institutions.

Reviewer 4 Report

Comments and Suggestions for Authors

In this study, HRV and heart-related biomarker data are applied to the deep neural network structure to perform severity diagnosis and subtype classification of mood disorders.

The study presents interesting results that have the potential to contribute to the related field. The article is well-written and follows the journal's rules. The manuscript is acceptable as is.

Author Response

Dear, Sensors reviewers.

First of all, thank you for suggesting us to submit revised draft of our manuscript entitled, "Mood disorder severity and subtype classification using Multimodal Deep Neural Network models" to Sensors. We also appreciate the time and effort you have dedicated to providing insightful feedback on ways to strengthen our paper. We thought it was a paper that lacked a lot, so our research team revised the entire manuscript to make a better paper. We have incorporated changes that reflect the detailed suggestions you have graciously provided. We also hope that our edits and the responses below satisfactorily address all the issues and concerns you have noted.

Response to Reviewer #4

Response: Our research team revised the overall manuscript to reflect the various opinions provided by the reviewers as much as possible. Various contents have been added, such as detailed explanations of the dataset, deep learning analysis algorithms, performance evaluation methods, and the meaning of this study in the future. We would really appreciate it if you could review this again.

Round 2

Reviewer 1 Report

Comments and Suggestions for Authors

Thanks for the corrections.

Reviewer 2 Report

Comments and Suggestions for Authors

Dear authors,

I suggest the acceptance in present form, thank you. 

Best regards,